# Cystoid Macular Edema after Rhegmatogenous Retinal Detachment Repair with Pars Plana Vitrectomy: Rate, Risk Factors, and Outcomes

**DOI:** 10.3390/jcm11164914

**Published:** 2022-08-21

**Authors:** Malik Merad, Fabien Vérité, Florian Baudin, Inès Ben Ghezala, Cyril Meillon, Alain Marie Bron, Louis Arnould, Pétra Eid, Catherine Creuzot-Garcher, Pierre-Henry Gabrielle

**Affiliations:** 1Department of Ophthalmology, Dijon University Hospital, 21000 Dijon, France; 2Agathe Group INSERM U 1150, UMR 7222 CNRS, ISIR (Institute of Intelligent Systems and Robotics), Sorbonne Université, 75005 Paris, France; 3Eye and Nutrition Research Group, Centre des Sciences du Goût et de l’Alimentation, AgroSup Dijon, CNRS, INRAE, Université de Bourgogne Franche-Comté, 21000 Dijon, France

**Keywords:** cystoid macular edema, rhegmatogenous retinal detachment, pars plana vitrectomy, vitreoretinal surgery, spectral-domain optical coherence tomography

## Abstract

(1) Background: The aim was to describe the rate and outcomes of cystoid macular edema (CME) after pars plana vitrectomy (PPV) for primary rhegmatogenous retinal detachment (RRD) and to identify risk factors and imaging characteristics. (2) Methods: A retrospective consecutive case study was conducted over a 5-year period among adult patients who underwent PPV for primary RRD repair. The main outcome measure was the rate of CME at 12 months following PPV. (3) Results: Overall, 493 eyes were included. The CME rate was 28% (93 patients) at 12 months. In multivariate analysis, eyes with worse presenting visual acuity (VA) (odds ratio [OR], 1.55; 95% CI, 1.07–2.25; *p* = 0.02) and grade C proliferative vitreoretinopathy (PVR) (OR, 2.88; 95% CI, 1.04–8.16; *p* = 0.04) were more at risk of developing CME 1 year after PPV. Endolaser retinopexy was associated with a greater risk of CME than cryotherapy retinopexy (OR, 3.06; 95% CI, 1.33–7.84; *p* = 0.01). Eyes undergoing cataract surgery within 6 months of the initial RRD repair were more likely to develop CME at 12 months (OR, 1.96; 95% CI, 1.06–3.63; *p* = 0.03). (4) Conclusions: CME is a common complication after PPV for primary RRD repair. Eyes with worse presenting VA, severe PVR at initial presentation, endolaser retinopexy, and cataract surgery within 6 months of initial RRD repair were risk factors for postoperative CME at 12 months.

## 1. Introduction

Rhegmatogenous retinal detachment (RRD) is a common and severe ocular condition affecting 6–18 patients per 100,000 each year [1,2]. Pars plana vitrectomy (PPV) has benefited from substantial technological advances in recent years, making this technique safe and effective for the treatment of RRD [3,4,5,6]. Functional recovery after RRD has now been widely studied, and cystoid macular edema (CME) is one of the main factors limiting functional recovery, along with epiretinal membrane formation (ERM) and photoreceptor alterations [7,8,9,10,11,12].

CME is a common retinal condition defined by an accumulation of fluid in the retina causing visual impairment [10,11,12,13,14,15]. CME is described in several ocular disorders, such as diabetic retinopathy or vascular occlusion, but it can also occur in postoperative situations, for example, cataract extraction or retinal detachment surgery [16,17]. Although large CMEs can be detected on fundus examination, retinal multimodal imaging provides a more refined diagnosis. The use of gold standard fluorescein angiography is continuously decreasing in favor of spectral-domain optical coherence tomography (SD-OCT). These major advances in noninvasive retinal imaging have enabled a better detection and follow-up of CME [18,19].

The rate of CME after primary RRD repair ranges from 6% to 36%, regardless of the surgical technique used [8,9,20,21,22]. Age, lens status (pseudophakia and aphakia), macular status, and the severity of the retinal detachment, especially with proliferative vitreoretinopathy (PVR), are reported to be risk factors for CME after primary RRD repair [9,20,23,24]. Recently, Pole et al. showed a relationship in univariate analysis between the number of surgeries, macular status, PVR grading, and the occurrence of CME, but these parameters were no longer significant in multivariate analysis [23]. There are still limited data on the rate, risk factors, and outcomes of CME after RRD repair with PPV [24,25]. Therefore, we aimed to report the rate of CME after PPV for primary RRD repair, identify the risk factors, and describe the 12-month outcomes and characteristics of SD-OCT imaging.

## 2. Materials and Methods

### 2.1. Design and Setting

Participants in this study were adult patients who underwent 23-gauge (G) or 25 G PPV for RRD primary repair at the Dijon University Hospital Ophthalmology Department between 1 January 2015 and 31 December 2019. This was a retrospective consecutive case study. All patients underwent either 23 G or 25 G primary PPV for RRD repair using the Constellation Vision System (Alcon Laboratories, Inc., Fort Worth, TX, USA) or Stellaris Vision Enhancement System (Bausch and Lomb, Inc., Rochester, NY, USA); a wide-angle viewing system was used for surgery. Surgeries were performed under general or local anesthesia induced by peribulbar block. All patients received postoperative treatment including 1 month of a topical nonsteroidal anti-inflammatory drug (NSAID), either indomethacin (Indocollyre^®^, Laboratoire Chauvin, Bausch and Lomb, Montpellier, France) or bromfenac (Yellox^®^, Croma Pharma GmbH, Korneuburg, Austria; and Bausch and Lomb, Inc., Rochester, NY, USA), 1 month of topical dexamethasone (Dexafree^®^, Laboratoires Théa, Clermont-Ferrand, France), 7 days of topical dorzolamide/timolol (Cosidime^®^, Laboratoire Santen, Paris, France), and 5 days of oral acetazolamide (Diamox^®^, Coopération Pharmaceutique Française, Melun, France). Ethical approval from an institutional review board was not required for this study due to its retrospective design, in accordance with French regulations. This study adhered to the tenets of the Declaration of Helsinki and followed the STROBE statements for reporting observational studies [26].

### 2.2. Data Sources and Measurements

Data were obtained retrospectively from electronic medical records, preoperatively and at 1, 3, 6, and 12 months postoperatively (M0, M1, M3, M6, and M12, respectively). Only one eye from the same patient was considered for this study. Baseline characteristics and preoperative patient data were obtained, including gender, age, diabetes mellitus, the affected eye, lens status, axial length, and intraocular pressure-lowering therapy. Visual acuity and characteristics of retinal detachment were recorded (extent of RRD (1–4 quadrants), number of retinal tears, presence of vitreous hemorrhage, PVR grade, macular status, onset of symptoms). The following surgery characteristics were retrieved: tamponade agent (air, hexafluoride (SF6), hexafluoroethane (C2F6), octafluoropropane (C3F8), silicone oil (SO)); retinopexy type (laser, cryotherapy or combined); the presence of retinotomy; retinectomy; internal limiting membrane peeling; use of perfluorocarbon liquid; and surgeon experience (fellow or senior surgeon). The time between RRD repair surgery and phacoemulsification was recorded for patients benefiting from sequential cataract surgery during the follow-up period.

During follow-up, we collected data on best-corrected visual acuity (BCVA) measurements in decimals, recurrence of RRD, and the number of surgeries. BCVA was converted to the logarithm of the minimum angle of resolution (LogMAR) and Snellen BCVA for statistical analysis. Single-surgery anatomical success (SSAS) was defined as retinal reattachment 12 months after a single operation. Final anatomical success (FAS) was defined as retinal reattachment at 12 months, requiring one or additional surgeries for recurrent retinal detachment [6].

Postoperative SD-OCT imaging was performed using the Cirrus high-definition SD-OCT system (Carl Zeiss, Dublin, CA, USA). CME diagnosis was defined as hyporeflective spaces regardless of the retinal layer involved. CRT was not considered for analysis because of the varying degree of macular atrophy [23]. SD-OCT characteristics were collected, such as the presence of ERM, central retinal thickness (CRT), outer retinal layer alterations (inner and outer segment junction (IS/OS) and/or external limiting membrane disruption), and localization of cysts in the inner retinal layers (IRL) or outer retinal layers (ORL).

### 2.3. Patient Selection and Groups

Current procedural terminology codes 67108 and 67113 were used to screen adult patients who underwent PPV for primary repair of retinal detachment between 1 January 2015 and 31 December 2019. Only one eye per patient was included. Exclusion criteria were: a follow-up period of fewer than 6 months, any history of RRD or vitreoretinal surgery (macular hole surgery, epiretinal membrane peeling, PPV), non-rhegmatogenous retinal detachment, severe ocular trauma, uveitis, endophthalmitis, and macular edema (age-related macular degeneration, diabetic macular edema, uveitis, retinal vein occlusion, a history of macular edema).

Patients presenting with CME at the 12-month follow-up were classified into all CME patients (aCME) and patients with no CME (nCME patients). Patients presenting with CME within 1 year at any visit (M1, M3, M6, and/or M12) were grouped into transient CME (tCME) and chronic CME (cCME), as described in previous studies [23,25]. tCME was defined as CME that appeared within the first 6 months of surgery, lasted less than 6 months during the follow-up period, and resolved spontaneously or with topical medications only. cCME was defined as CME seen on OCT imaging at least 6 months apart and/or benefiting from more complex CME management with additional treatment such as systemic medication, intravitreal injection of dexamethasone implant, and/or surgery with ERM peeling. The management of CME was based on symptoms, visual acuity, and multimodal imaging at the discretion of the physician in consultation with the patient, thereby representing routine clinical practice.

### 2.4. Outcomes

The main outcome was the rate of CME at 12 months following primary RRD repair with PPV. We also examined risk factors for CME at 12 months and postoperative CME SD-OCT characteristics, SSAS rate, FAS rate, and visual outcomes at 12 months.

### 2.5. Statistical Analysis

Continuous variables are described as the median and interquartile range (IQR) or mean and standard deviation (SD) according to their distribution, while categorical variables are given by number and percentage. Risk factors for CME at 12 months were assessed through univariate and multivariate logistic regressions. Factors with a significance level lower than 20% in univariate regression and with less than 5% of missing values were retained for multivariate regressions. Results of regressions are expressed as odds ratios (ORs) with 95% confidence interval and *p* values. Risk factors for chronic macular edema on SD-OCT imaging were assessed by univariate logistic regression. Missing values are reported in tables and were excluded from the analysis. The significance threshold was 5%. Boxplots were used to display the evolution of BCVA and CRT after surgery. All analyses were conducted using R Statistical Software version 4.0.1 (R Foundation for Statistical Computing, Vienna, Austria, 2021) with the gtsummary package (V 1.4.1).

## 3. Results

### 3.1. Study Participants

A total of 1042 patients underwent PPV surgery for primary retinal detachment repair between 1 January 2015 and 31 December 2019. Of these patients, 493 met the inclusion criteria (Figure 1). Baseline clinical and demographic characteristics are described in Table 1. Most patients were male (66%), the mean age was 63.0 (±10.9) years, and the mean BCVA at initial presentation was 1.23 (±0.93) LogMAR (20/400 Snellen VA chart).

### 3.2. Cystoid Macular Edema Rate and Risk Factors

Overall, 165 (34%, six missing data) patients experienced at least one episode of CME within the first year of follow-up. The rate of CME at 12 months was 28% (97 patients, 148 missing data). Baseline demographics and surgery characteristics are shown in Table 2 and Table 3 for univariate and multivariate analysis, respectively. Due to missing data at 12 months, 345 patients were included in the univariate analysis and 327 in the multivariate analysis for risk factors of CME at 12 months. aCME and nCME patients were similar in age (*p* = 0.86), gender (*p* = 0.52), and axial length (*p* = 0.73) at baseline.

In univariate analysis, risk factors for CME at 12 months were lower initial BCVA (OR, 1.72; 95% CI, 1.32–2.27; *p* < 0.001), macula-off status (OR, 1.80; 95% CI, 1.02–3.28; *p* = 0.048), greater extent of RRD (OR, 1.43; 95% CI, 1.13–1.81; *p* = 0.003), severe grade C PVR (OR, 4.01; 95% CI, 2.22–7.28; *p* < 0.001), and SO tamponade (OR, 2.84; 95% CI, 1.57–5.13; *p* < 0.001). Eyes with early recurrence of RRD (within 3 months) and increased number of surgeries were at a greater risk of postoperative CME. Surgeon experience was not statistically associated with CME at 12 months (*p* = 0.57), nor were lens status (*p* = 0.88), vitreous hemorrhage at presentation (*p* = 0.54), and diabetes mellitus (*p* = 0.67).

In multivariate analysis, eyes with worse initial BCVA (OR, 1.55; 95% CI, 1.07–2.25; *p* = 0.02) and severe grade C PVR (OR, 2.88; 95% CI, 1.04–8.16; *p* = 0.04) had an increased risk of postoperative CME at 12 months (Table 3). Endolaser retinopexy was associated with an increased risk for postoperative CME at 12 months when compared to cryotherapy retinopexy (OR, 3.06; 95% CI, 1.33–7.84; *p* = 0.01). Post-vitrectomy cataract surgery within 6 months of the initial RRD repair was associated with CME at 12 months (OR, 1.96; 95% CI, 1.06–3.63; *p* = 0.03).

### 3.3. Visual and Anatomical Outcomes

SSAS was achieved in 370 (75%) patients, while the FAS rate was 98% (*n* = 479) at 12 months. A total of 116 patients experienced re-detachment during the study period, with 85 of these cases (73%) occurring within 3 months of initial surgery. An epiretinal membrane was found in 132 (38%) patients at 12 months. Median CRT was 282.5 (249.0–321.0) µm at 12 months. BCVA increased significantly after surgery until 1 month and remained stable until 12 months, with a median final BCVA of 0.22 (0.05–0.75) LogMAR (20/32 Snellen VA chart). Overall, 61 (16%) patients had a final BCVA of <20/200, 316 (84%) patients had a final BCVA of ≥20/200, and 209 (55%) patients achieved a final BCVA of ≥20/40.

During the study period, 156 (32%) patients received topical NSAIDs, 100 (20%) patients received oral acetazolamide. Dexamethasone injections (Ozurdex^®^, Allergan Pharmaceuticals Ireland, Dublin, Ireland) were used for 19 (4%) patients, and 17 (4%) patients underwent surgery for ERM peeling.

### 3.4. SD-OCT Characteristics of Postoperative CME

The CME characteristics were assessed using SD-OCT imaging for 161 patients and are shown in Table 4. Fifty-six patients were classified as having tCME, and 105 patients were classified as having cCME. Outer retinal layer alterations were present in 124 (77%) patients and more frequently in cCME patients (*p* = 0.046). Subretinal fluid was present in 15 (9%) patients and did not differ significantly between the tCME and cCME groups. CME localization in both the IRL and ORL (Figure 2) was a risk factor for cCME (OR, 6.44; 95% CI, 3.08–14.3; *p* < 0.001) in univariate analysis when compared to macular cysts located in the IRL only (Figure 3).

## 4. Discussion

This study evaluated the rate of CME 1 year after PPV for primary RRD repair in routine clinical practice. Overall, 34% of patients (*n* = 163) experienced at least one episode of CME within the first year of follow-up, while the rate of CME at 12 months was 28% (97 patients). When considering a CRT greater than 320 µm associated with macular cysts in SD-OCT for the diagnosis of CME, the rate of CME at 12 months decreased to 13% [27]. These findings are consistent with previous studies, although the range of CME rates varies greatly, primarily due to a difference in CME diagnostic criteria [27,28]. A recent single-center retrospective study with a smaller sample size but similar CME diagnostic criteria showed a CME rate of 25% among patients who underwent surgery with PPV or combined PPV and scleral buckling. Lower rates were previously reported, which may be explained by discrepancies in selection criteria with the exclusion of severe cases of RRD (PVR, SO tamponade, high myopia) or in follow-up time [8,22,24]. A study conducted by Yang et al. found a higher rate (36%) of postoperative CME, probably due to the severity of RRD in the eyes included in this study (SO tamponade only) [20]. Our rate might be overestimated owing to missing data (*n* = 148, 30%) at 12 months, though no imputations were carried out, and patients with good anatomical and functional outcomes were more prone to resuming their follow-up outside of our academic institution.

Several risk factors for CME were described, such as age, pseudophakia and aphakia, macular status, severe PVR grade, retinectomy, number of surgeries, and RRD duration [9,23,24,28]. We found that eyes with worse BCVA and severe grade C PVR at baseline, and eyes that had endolaser retinopexy and cataract surgery within 6 months of primary RRD repair, were more likely to have CME 1 year after the primary repair. Severe PVR, lower initial BCVA, greater RRD extent, SO tamponade, recurrence within 3 months, retinectomy, and internal limiting membrane peeling are all RRD characteristics suggesting initial severity. They were associated with CME in univariate analysis, but only severe grade C PVR and initial BCVA remained significantly associated with CME at 12 months in the multivariate analysis. These findings confirm that inflammation is the leading process involved in CME development after RRD since PVR membranes form in response to a major increase in inflammatory cytokines following tissue damage caused by RRD and resultant inflammation. The use of endolaser retinopexy was associated with a significantly higher risk of postoperative CME than cryotherapy retinopexy, which was not previously described in earlier studies. In our institution, we believe that the initial severity of the RRD was a determinant of the retinopexy used, with cryotherapy mostly used for less severe or single-tear RRDs, which could explain the higher rate of CME in the endolaser retinopexy group. Moreover, 40% of patients had extensive endolaser retinopexy (360 degrees encircling) in the aCME group versus only 24% in the nCME group, which might increase the risk of ERM formation and CME occurrence in this group. Unfortunately, data on the average endolaser power and the total number of delivered spots were unavailable. Moreover, laser retinopexy was not performed in a standardized fashion in our study with several surgeons.

Lens status at baseline was not associated with CME at 12 months (*p* = 0.88). Lens status remains a controversial risk factor for CME in the literature. Meredith et al. and Chatziralli et al. found that pseudophakia was a risk factor for CME, although it was not confirmed in other studies [9,23,24,27,28]. Secondary cataract surgery within 6 months of initial surgery was not significantly associated with CME in univariate analysis, but the association was significant in multivariate analysis. Further analyses are needed to assess whether some patients could benefit from a postponed cataract surgery after RRD repair surgery.

Several studies investigated SD-OCT macular characteristics after RRD repair. Photoreceptor layer alterations are common and are associated with poor functional outcomes [8,11,27]. Similarly, CME was reported to alter BCVA in patients with a history of RRD [10]. Among patients presenting with CME, macular cysts were mainly localized in the IRL and, more precisely, in the INL. The distribution of macular cysts in both the IRL and ORL was associated with cCME. These findings are consistent with recent studies suggesting that cCME shares features with uveitic CME, whereas tCME might be a variant of pseudophakic CME [18,23,29].

Surgeon experience was not significantly associated with CME at 12 months in both univariate and multivariate analyses even though the SSAS rate was lower and more surgeries were needed to achieve FAS in the fellow surgeon group than the experienced surgeon group. The number of surgeries tended to influence CME formation but did not reach statistical significance in multivariate analysis (*p* = 0.06), as previously described [23]. The SSAS rate in our study (75%) at 12 months was in the lower range compared to previous studies (82–100%) [4,30,31,32,33]. As expected in a tertiary referral center, a majority of complex RRD cases were included. Indeed, 72% of patients had detached macula, 26% had RRD involving the four quadrants, and 15% had severe PVR (grade C). Almost half of the surgical procedures (49%) were performed by fellow vitreoretinal surgeons. Nevertheless, the FAS rate was high at the end of the follow-up period, showing the effectiveness of PPV for RRD repair.

Our study had several strengths. First, this real-world study had a large sample size with broad inclusion criteria incorporating complex RRD with severe PVR and SO tamponade, as well as a long follow-up period of 12 months. Anatomical and functional outcomes are within the range of previous studies in terms of CME, SSAS, and FAS rates. We acknowledged some weaknesses inherent in retrospective studies. The diagnosis of CME was based on the presence of hyporeflective space on SD-OCT imaging without dye angiography characteristics. Thus, the rate of CME might include both inflammatory CME with intraretinal leakage on dye angiography and macular cystoid degeneration without intraretinal leakage on dye angiography [34]. Nonetheless, CME after RRD repair may have distinct pathophysiology from other etiologies of CME [35]. Our department is a tertiary referral center, and more complex RRD cases may be referred to us. Some patients were referred to their ophthalmologists before the 6-month follow-up and were not included in this study. We also had missing data at 12 months, reducing our sample size for the analysis of risk factors. The study design did not allow us to use a Cox regression model to fix the onset of CME and adapt the monitoring of patients. Cataract formation or cataract surgery during the follow-up period might have influenced the visual outcomes. However, the final BCVA should not have been significantly impacted since, among our 304 phakic patients, only 30 were still phakic at the end of the follow-up period. Finally, the management of CME was made without reference to a guided protocol and is likely to differ among physicians (based on the anatomical and functional outcome of the surgery), which may have influenced the rate of CME at 12 months. Unfortunately, the study design did not allow us to assess and compare the outcome of a given treatment during the study period.

In conclusion, CME was a common complication 1 year after PPV for primary RRD repair in routine clinical practice. It might be helpful to address this matter with patients when explaining the surgical procedure, as CME might lead to repeated medical visits and chronic treatments. Localization of macular cysts in both the IRL and ORL on SD-OCT seemed to be a useful OCT biomarker of postoperative CME chronicity following RRD repair. Patients with low presenting VA and severe PVR should be monitored carefully for CME after RRD repair. Similarly, patients who had endolaser retinopexy and underwent cataract surgery within 6 months of RRD repair were more likely to have CME at 12 months. Further research is needed to determine whether patients may benefit from delayed cataract surgery.

## Figures and Tables

**Figure 1 jcm-11-04914-f001:**
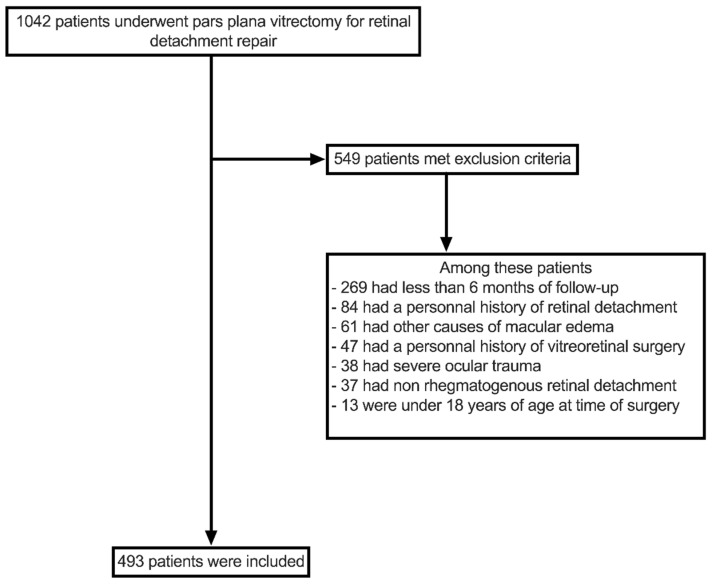
Flowchart of patient selection process.

**Figure 2 jcm-11-04914-f002:**
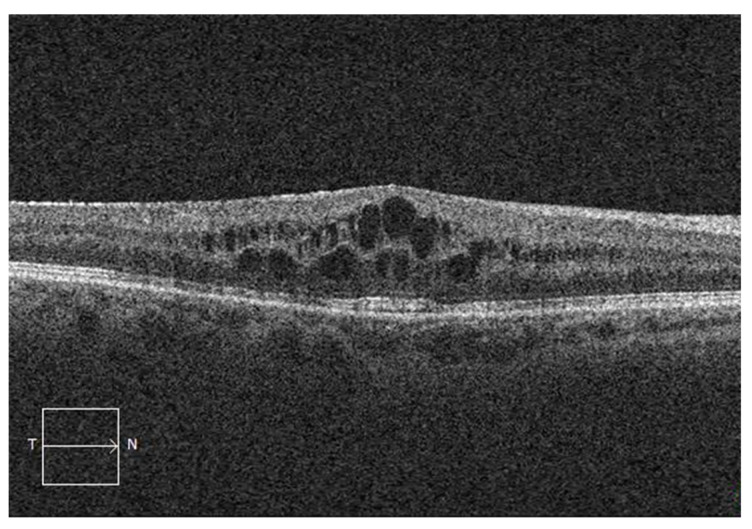
Spectral-domain optical coherence tomography showing cystoid macular edema in both inner and outer retinal layers after vitrectomy for rhegmatogenous retinal detachment.

**Figure 3 jcm-11-04914-f003:**
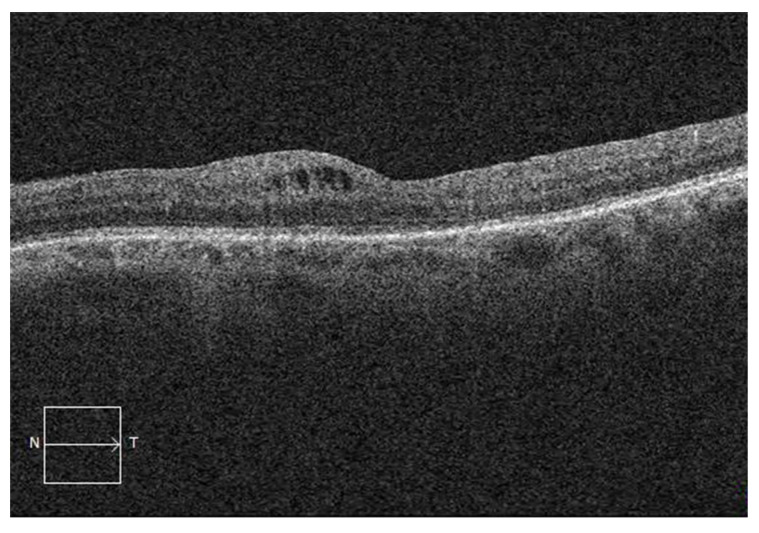
Spectral-domain optical coherence tomography showing cystoid macular edema in the inner retinal layers only after vitrectomy for rhegmatogenous retinal detachment.

**Table 1 jcm-11-04914-t001:** Population description.

	*n* = 493
**Age (year)**	63 (10.9)
**Gender-Male**	324 (65.7%)
**Lens status**	
*Phakic*	304 (61.7%)
*Pseudophakic*	187 (37.9%)
*Aphakic*	2 (0.4%)
**Axial length (mm) ***	24.78 (1.78)
**Diabetes mellitus ***	43 (8.78%)
**Initial BCVA (LogMAR) ***	1.23 (0.93)
**Extent of RRD (1–4 quadrants)**	
*1*	66 (13.4%)
*2*	209 (42.4%)
*3*	90 (18.2%)
*4*	128 (26.0%)
**Number of retinal tears ***	
*0*	23 (4.7%)
*1*	253 (51.7%)
*2*	101 (20.7%)
*3*	57 (11.7%)
*4 and more*	55 (11.2%)
**Proliferative vitreoretinopathy ***	
*Grade A*	101 (21.0%)
*Grade B*	305 (63.6%)
*Grade C*	74 (15.4%)
**Macular status**	
*On*	138 (28.0%)
*Off*	355 (72.0%)
**Time to surgery after first symptoms (days) ***	17 (22)
**Surgical procedure**	
*Pseudophakic-PPV only*	182 (36.9%)
*Phakic-PPV only*	258 (52.3%)
*Phakic-PPV with simultaneous PKE*	53 (10.8%)
**Secondary cataract surgery**	
*Not applicable*	272 (55.2%)
*Under 3 months*	51 (10.3%)
*Between 3 and 6 months*	81 (16.4%)
*Between 6 and 12 months*	89 (18.1%)
**Single surgery anatomical success**	370 (75.0%)
Cystoid macular edema rate within the first 12 months *^,†^	165 (33.9%)
**Cystoid macular edema rate at 12 months ***	97 (28.1%)

BCVA, best-corrected visual acuity; RRD, rhegmatogenous retinal detachment; PPV, pars plana vitrectomy; PKE, phacoemulsification; M6, 6 months after RRD repair; M12, 12 months after RRD repair. Categorical variables are described as *n* (%); continuous variables are described as mean (SD). * Missing data (*n*): axial length (153); diabetes mellitus (3); initial BCVA (11); number of retinal tears (4); proliferative vitreoretinopathy (13); time to surgery after first symptoms (102); cystoid macular edema rate within the first 12 months (6); cystoid macular edema rate at M12 (148). ^†^ Patients that experienced at least one episode of CME at any visits within the first year.

**Table 2 jcm-11-04914-t002:** Risk factors for cystoid macular edema at 12 months univariate analysis.

	nCME (*n* = 248)	aCME (*n* = 97)	OR *	95% CI	*p* Value
**Age (year)**	62 (56, 69)	65 (57, 70)	1.00	0.98–1.02	0.86
**Gender-Male**	165 (66.5%)	61 (62.9%)	0.85	0.52–1.40	0.52
**Lens status**					
*Phakic*	155 (62.5%)	60 (61.9%)	—	—	
*Pseudophakic*	92 (37.1%)	37 (38.1%)	1.04	0.64–1.68	0.88
*Aphakic*	1 (0.4%)	0 (0.0%)	0.00		>0.99
**Axial length (mm) ^†^**	24.44 (23.68, 25.70)	24.64 (23.95, 25.65)	1.03	0.88–1.20	0.73
**Diabetes mellitus ^†^**	22 (8.9%)	10 (10.4%)	1.19	0.52–2.55	0.67
**Initial BCVA (LogMAR) ^†^**	0.85 (0.16, 2.30)	2.30 (0.70, 2.30)	1.72	1.32–2.27	**<0.001**
**Macular status-Off**	175 (71.0%)	79 (81.4%)	1.80	1.02–3.28	**0.048**
**Extent of RRD (1–4 quadrants)**			1.43	1.13–1.81	**0.003**
*1*	36 (14.5%)	7 (7.2%)			
*2*	107 (43.1%)	36 (37.1%)			
*3*	49 (19.8%)	17 (17.6%)			
*4*	56 (22.6%)	37 (38.1%)			
**Proliferative vitreoretinopathy ^†^**					
*Grade A and B*	215 (89.2%)	64 (67.4%)	—	—	
*Grade C*	26 (10.8%)	31 (32.6%)	4.01	2.22–7.28	**<0.001**
**Vitreous hemorrhage**	37 (14.9%)	12 (12.4%)	0.81	0.39–1.58	0.54
**Time to surgery after first symptoms (days) ^†^**	10 (5, 20)	8 (5, 14)	1.01	1.00–1.02	0.19
**Surgical procedure**					
*Phakic-PPV only*	136 (54.8%)	49 (50.5%)	—	—	
*Pseudophakic-PPV only*	92 (37.1%)	36 (37.1%)	1.09	0.65–1.80	0.75
*Phakic-PPV with simultaneous PKE*	20 (8.1%)	12 (12.4%)	1.67	0.74–3.62	0.20
**Tamponade agent**					
*Hexafluoride (SF6)*	161 (64.9%)	47 (48.5%)	—	—	
*Air*	2 (0.8%)	0 (0.0%)	0.00		0.98
*Hexafluoroethane (C2F6)*	45 (18.2%)	20 (20.6%)	1.52	0.81–2.81	0.18
*Octafluoropropane (C3F8)*	5 (2.0%)	1 (1.0%)	0.69	0.04–4.39	0.73
*Silicone oil*	35 (14.1%)	29 (29.9%)	2.84	1.57–5.13	**<0.001**
**Retinopexy type ^†^**					
*Cryotherapy*	65 (26.4%)	9 (9.3%)	—	—	
*Endolaser*	135 (54.9%)	75 (77.3%)	4.01	1.98–9.05	**<0.001**
*Combined cryotherapy and endolaser*	46 (18.7%)	13 (13.4%)	2.04	0.81–5.33	0.13
**Retinotomy ^†^**	66 (26.7%)	28 (28.9%)	1.11	0.65–1.86	0.69
**Retinectomy ^†^**	2 (0.8%)	5 (5.2%)	6.66	1.41–47.1	**0.02**
**Internal limiting membrane peeling ^†^**	27 (10.9%)	26 (27.1%)	3.03	1.65–5.54	**<0.001**
**Use of PFCL ^†^**	38 (15.5%)	34 (35.4%)	3.00	1.74–5.17	**<0.001**
**Surgeon experience-fellow surgeon**	122 (49.2%)	51 (52.6%)	1.15	0.72–1.84	0.57
**Secondary cataract surgery within 6 months**	64 (25.8%)	34 (35.1%)	1.55	0.93–2.56	0.09
**Single surgery anatomical success**	194 (78.2%)	60 (61.9%)	2.22	1.33–3.68	**0.002**
**Number of RRD repair within 12 months ^†^**			1.88	1.38–2.60	**<0.001**
*1*	201 (81.1%)	59 (61.5%)			
*2*	33 (13.3%)	21 (21.9%)			
*3*	11 (4.4%)	10 (10.4%)			
*4*	3 (1.2%)	6 (6.2%)			
**Retinal re-detachment within 3 months ^†^**	34 (13.7%)	28 (29.2%)	2.59	1.46–4.58	**0.001**

OR, odds ratio; CI, confidence interval; nCME, no cystoid macular edema group; aCME, cystoid macular edema group, BCVA, best-corrected visual acuity; RRD, rhegmatogenous retinal detachment; PPV, pars plana vitrectomy; PKE, phacoemulsification; PFCL, perfluorocarbon liquid. Categorical variables are described as *n* (%), continuous variables are described as median (IQR). * Estimated from univariate logistic regression model. ^†^ Missing data (nCME; aCME): axial length (74; 28); diabetes mellitus (1; 1); initial BCVA (5; 2); proliferative vitreoretinopathy (7; 2); time to surgery after first symptoms (55; 17); retinopexy type (2; 0); retinotomy (1; 0); retinectomy (1; 0); internal limiting membrane peeling (1; 1); use of PCFL (2; 1); number of RRD repair within 12 months (0; 1); retinal re-detachment within 3 months (0; 1).

**Table 3 jcm-11-04914-t003:** Risk factors for cystoid macular edema at 12 months-multivariate analysis.

	nCME (*n* = 236)	aCME (*n* = 91)	OR *	95% CI	*p* Value
**Initial BCVA (LogMAR)**	0.85 (0.16, 2.30)	2.30 (0.81, 2.30)	1.55	1.07–2.25	**0.02**
**Macular status-Off**	168 (71.2%)	77 (84.6%)	1.16	0.51–2.71	0.72
**Extent of RRD (1–4 quadrants)**			1.01	0.72–1.42	0.95
*1*	32 (13.6%)	5 (5.5%)			
*2*	103 (43.6%)	34 (37.4%)			
*3*	46 (19.5%)	15 (16.5%)			
*4*	55 (23.3%)	37 (40.6%)			
**Proliferative vitreoretinopathy**					
*Grade A and B*	211 (89.4%)	62 (68.1%)	—	—	
*Grade C*	25 (10.6%)	29 (31.9%)	2.88	1.04–8.16	**0.04**
**Tamponade agent**					
*Hexafluoride (SF6)*	153 (64.8%)	45 (49.4%)	—	—	
*Air*	2 (0.9%)	0 (0.0%)	0.00		0.99
*Hexafluoroethane (C2F6)*	43 (18.2%)	17 (18.7%)	0.80	0.36–1.70	0.57
*Octafluoropropane (C3F8)*	5 (2.1%)	1 (1.1%)	0.29	0.01–2.44	0.32
*Silicone oil*	33 (14.0%)	28 (30.8%)	0.65	0.22–1.78	0.41
**Retinopexy type**					
*Cryotherapy*	63 (26,7%)	8 (8.8%)	—	—	
*Endolaser*	128 (54.2%)	70 (76.9%)	3.06	1.33–7.84	**0.01**
*Combined cryotherapy and endolaser*	45 (19.1%)	13 (14.3%)	1.46	0.52–4.23	0.47
**Retinectomy**	2 (0.9%)	4 (4.4%)	1.86	0.31–15.0	0.51
**Internal limiting membrane peeling**	26 (11.0%)	26 (28.6%)	0.97	0.36–2.51	0.94
**Use of PFCL**	37 (15.7%)	33 (36.3%)	1.43	0.60–3.35	0.41
**Secondary cataract surgery within 6 months**	59 (25.0%)	31 (34.1%)	1.96	1.06–3.63	**0.03**
**Single surgery anatomical success**	189 (80.1%)	57 (62.6%)	0.67	0.18–2.15	0.51
**Number of RRD repair within 12 months**			2.09	1.00–4.67	0.06
*1*	196 (83.0%)	57 (62.6%)			
*2*	29 (12.3%)	21 (23.1%)			
*3*	9 (3.8%)	8 (8.8%)			
*4*	2 (0.9%)	5 (5.5%)			
**Retinal re-detachment within 3 months**	28 (11.9%)	25 (27.5%)	1.18	0.39–3.67	0.77

OR, odds ratio; CI, confidence interval; nCME, no cystoid macular edema group; aCME, cystoid macular edema group; BCVA, best-corrected visual acuity; RRD, rhegmatogenous retinal detachment; PFCL, perfluorocarbon liquid. Categorical variables are described as *n* (%), continuous variables are described as median (IQR). * Estimated from mutlivariate logistic regression model.

**Table 4 jcm-11-04914-t004:** Cystoid macular edema characteristics in SD-OCT imaging.

	tCME (*n* = 56)	cCME (*n* = 105)	OR *	95% CI	*p* Value
**Outer retinal layer alteration**	38 (67.9%)	86 (81.9%)	2.14	1.01–4.56	**0.046**
**Macular cysts localization**					
*IRL*	41 (73.2%)	35 (33.3%)			
*ORL*	1 (1.8%)	4 (3.8%)	4.69	0.66–94.0	0.18
*IRL and ORL*	12 (21.4%)	66 (62.9%)	6.44	3.08–14.3	**<0.001**
**Subretinal fluid**	3 (5.4%)	12 (11.4%)	2.28	0.69–10.3	0.22

SD-OCT, spectral-domain optical coherence tomography; tCME, transient cystoid macular edema group; cCME, chronic cystoid macular edema group; OR, odds ratio; CI, confidence interval; IRL, inner retinal layers; ORL, outer retinal layers. Categorical variables are described as *n* (%). * Estimated from univariate logistic regression model.

## Data Availability

Not applicable.

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
