# Peer review of "Cystoid Macular Edema after Rhegmatogenous Retinal Detachment Repair with Pars Plana Vitrectomy: Rate, Risk Factors, and Outcomes"

_jcm, 2022, doi:10.3390/jcm11164914_

Round 1
Reviewer 1 Report
In this paper, the authors set out to determine the risk factors, imaging characteristics, rate and outcomes of CME after PPV for RRD repair. They suggest that 28% of patients had CME 12 months after PPV, and risk factors for developing CME were worse presenting vision, grade C PVR, endolaser exposure, and undergoing cataract surgery within 6 months. They conclude that CME is common after primary RRD repair.
Observations and issues needing to be addressed:
- It may be useful to report diabetic status. Even if patients with diabetic macular edema were excluded, the presence of diabetes may predispose patients to developing macular edema after intraocular surgery. What was the timing of CME in relation to cataract surgery?
- Were any patients on prostaglandin analogues?
- It is interesting that all patients received topical NSAID, dexamethasone, dorzolamide/timolol, and oral acetazolamide. Please clarify reasoning for placing all patients on dorzolamide /timolol and acetazolamide.
- It is interesting that the patients with endolaser exposure were more at risk for developing CME since cryotherapy is more pro-inflammatory. Please comment on proposed explanation for this observation. Was there selection bias in that those with more extensive pathology received more laser? Is it possible that those who underwent cryopexy eventually redetached and were excluded from the study?
- It is unclear how the results of this study would influence the management of retinal detachments and how it can be applied to patient care. Please elaborate. It may be more useful to assess the outcome of a given treatment.
- It would also be useful to report timing of resolution of CME based on intervention/treatment.
- Of the patients who had transient CME, what percentage of these patients were treated with additional topical medication vs injections? Did treatment affect resolution time or visual acuity improvement? Were any adjuncts used during surgery such as triamcinolone to aid in visualization of the vitreous base?
- Of the patients who had chronic CME, what percentage of these patients were treated with additional topical medication vs injections? Did treatment affect resolution time or visual acuity improvement?
Author Response
Thank you for allowing us to revise this submission following peer review and for this thorough and helpful analysis of our research work. After consultation with the other authors, we reviewed the manuscript and performed the suggested improvements. Our point-by-point response to the Reviewer’s comments, suggestions, and questions are below.
Reviewer 1
In this paper, the authors set out to determine the risk factors, imaging characteristics, rate and outcomes of CME after PPV for RRD repair. They suggest that 28% of patients had CME 12 months after PPV, and risk factors for developing CME were worse presenting vision, grade C PVR, endolaser exposure, and undergoing cataract surgery within 6 months. They conclude that CME is common after primary RRD repair.
We thank Reviewer 1 for these encouraging comments and recommendations in improving our manuscript.
Observations and issues needing to be addressed:
- It may be useful to report diabetic status. Even if patients with diabetic macular edema were excluded, the presence of diabetes may predispose patients to developing macular edema after intraocular surgery. What was the timing of CME in relation to cataract surgery?
We would like to thank the Reviewer for raising this interesting point. Even though patients with diabetic macular edema were excluded, some patients had diabetes Mellitus. In our study population (493 patients), 43 patients had diabetes (8,78%). Diabetic status was included in the univariate analysis exploring risk factors for CME. It was not associated with an increased incidence of CME in our population (P = 0.67).
The manuscript was modified accordingly:
Page 4-5. Results section, Table 1 population description.
Page 6, line 175: “Surgeon experience was not statistically associated with CME at 12 months (P = 0.57), nor were lens status (P = 0.88), vitreous hemorrhage at presentation (P = 0.54) and diabetes mellitus (P = 0.67).”
Page 6. Results section, Table 2 Risk factors for cystoid macular edema at 12 months – univariate analysis.
The precise date of cataract surgery was not collected in regard to the CME status of the patients; thus, we were unable to evaluate its influence on the timing of CME and visual acuity.
- Were any patients on prostaglandin analogues?
In our study population (493 patients), only 18 patients were treated with prostaglandin analogs at initial presentation (4 missing data). We, therefore, chose not to include this data in the statistical analyses.
- It is interesting that all patients received topical NSAID, dexamethasone, dorzolamide/timolol, and oral acetazolamide. Please clarify reasoning for placing all patients on dorzolamide /timolol and acetazolamide.
The reason behind this treatment is due to our local institutional management for patients undergoing PPV for RRD with either gas or silicon oil as a tamponade agent. Our institution's first follow-up exam is performed approximately seven days after the surgery. This treatment is given to prevent ocular hypertension during the first postoperative week.
- It is interesting that the patients with endolaser exposure were more at risk for developing CME since cryotherapy is more pro-inflammatory. Please comment on proposed explanation for this observation. Was there selection bias in that those with more extensive pathology received more laser?
Many thanks to the Reviewer for pointing this out. Indeed, we found in our study that patients with endolaser were more at risk of developing CME at 12 months in univariate analysis, which was confirmed in multivariate analysis. Yes, a selection bias is present due to the use of endolaser in case of extensive retinal detachment or multiple retinal tears. These factors are related to the initial severity of the RRD, which lead to an increased risk of developing CME, as explained in page 10, lines 244-248 of the revised manuscript.
“The use of endolaser retinopexy was associated with a significantly higher risk of postoperative CME than cryotherapy retinopexy, which has not been previously described in earlier studies. Cryotherapy is mostly used for single-tear RRDs, which could be a confounding factor for more severe RRDs in the endolaser retinopexy group.”
- Is it possible that those who underwent cryopexy eventually redetached and were excluded from the study?
Patients presenting a retinal re-detachment during the follow-up were not excluded from the analysis.
- It is unclear how the results of this study would influence the management of retinal detachments and how it can be applied to patient care. Please elaborate. It may be more useful to assess the outcome of a given treatment.
We found that CME is common after primary RRD repair. The primary objective was to determine the rate of CME after PPV since few authors have addressed this matter after PPV only but rather after either scleral buckling or PPV.
We also aimed to find risk factors for CME. Our study suggests that a greater initial severity (grade C PVR, worse initial visual acuity) is a risk factor for developing CME.
Consequently, we think this study might help physicians to give adequate explanations to their patients depending on the severity of the RRD at initial presentation. We usually highlight the anatomical result, including the rate of reapplication without recurrence. However, it may be beneficial to emphasize the potential complications such as CME that may lead to repeated medical visits and chronic treatments.
We added the following remark in our revised manuscript:
Page 11, lines 303-305: “It might be helpful to address this matter with patients when explaining the surgical procedure, as CME might lead to repeated medical visits and chronic treatments.”
- It would also be useful to report timing of resolution of CME based on intervention/treatment.
- Of the patients who had transient CME, what percentage of these patients were treated with additional topical medication vs injections? Did treatment affect resolution time or visual acuity improvement? Were any adjuncts used during surgery such as triamcinolone to aid in visualization of the vitreous base?
- Of the patients who had chronic CME, what percentage of these patients were treated with additional topical medication vs injections? Did treatment affect resolution time or visual acuity improvement?
We would like to thank the Reviewer for raising these interesting comments regarding intervention and resolution of CME. Unfortunately, the study's design did not allow us to draw conclusions on CME resolution with received treatments. Due to the retrospective design, the management of CME was left at the surgeon’s discretion. Therefore, the decision to treat each individual could be different depending on whether physicians considered the CME to be clinically significant based on the anatomical and functional outcome of the surgery. Moreover, no standardized treatment or monitoring protocol was applied.
In addition, we rarely use triamcinolone as an adjunct during surgery, except in some highly myopic patients. In case of epiretinal membrane, we usually use dual blue membrane for membrane and ILM peeling.
We suggest that further studies be conducted to determine and compare the effect of NSAIDs, steroids, carbonic anhydrase inhibitors, and vitrectomy or ERM peeling in this case.
The manuscript was modified accordingly:
Page 11, lines 296-300: “Finally, the management of CME was made without reference to a guided protocol and is likely to differ among physicians (based on the anatomical and functional outcome of the surgery), which may have influenced the rate of CME at 12 months. Unfortunately, the study design did not allow us to assess and compare the outcome of a given treatment during the study period.”
Reviewer 2 Report
A well designed study with clear methodology and results.
A confounding factor for BCVA determination post VR surgery would be cataract formation, and cataract surgery. So these factors should be taken into account while analysing BCVA improvement.
Also, incidence of CME following cryotherapy is generally more than that seen after endolaser. Interestingly this study finds the reverse. But as the authors have explained, that in their series, the cases in which cryopexy was done were those with limited area of lesions requiring limited retinopexy, while endolaser was used in cases with more extensive lesions or ones requiring 360 degrees retinopexy. A categorization of lesions based on extent and nature viz-a-viz standardization of treatment protocol, specifically cryopexy Vs endolaser could have given us a clearer picture.
Author Response
Reviewer 2
- A well designed study with clear methodology and results.
We thank Reviewer 2 for these encouraging comments and recommendations in improving our manuscript.
- A confounding factor for BCVA determination post VR surgery would be cataract formation, and cataract surgery. So these factors should be taken into account while analysing BCVA improvement.
The precise date of cataract surgery was not collected in regard to the CME status of the patients; thus, we were unable to evaluate its influence on the timing of CME and visual acuity.
However, among the 304 phakic patients in our population, 53 had PPV with simultaneous phakoemulsification, and 221 had their lens removed during the first year. This leaves only about 30 phakic patients at the end of the follow-up period. Therefore, while visual acuity and CME occurrence during the follow-up period might have been influenced by cataract formation and surgery, final visual acuity at 12 months should not have been significantly impacted.
- Also, incidence of CME following cryotherapy is generally more than that seen after endolaser. Interestingly this study finds the reverse. But as the authors have explained, that in their series, the cases in which cryopexy was done were those with limited area of lesions requiring limited retinopexy, while endolaser was used in cases with more extensive lesions or ones requiring 360 degrees retinopexy. A categorization of lesions based on extent and nature viz-a-viz standardization of treatment protocol, specifically cryopexy Vs endolaser could have given us a clearer picture.
Thank you for raising this interesting point. This result was not found in previous studies.
We found in our study that patients with endolaser were more at risk of developing CME at 12 months in univariate analysis, which was confirmed in multivariate analysis. Indeed, we believe that the initial severity was a determinant of the retinopexy used with cryopexy used for less severe retinal detachment. This might explain the higher rate of CME in the laser retinopexy group.
It would be interesting to investigate the characteristics of RD leading to the choice of either cryotherapy or laser retinopexy. Surgeons’ habits and preferences should also be taken into account.